# Reliable Reasoning Beyond Natural Language

## Abstract

Despite their linguistic competence, Large Language Models (LLMs) often struggle to reason reliably and flexibly. To identify these shortcomings, we introduce the Non-Linear Reasoning (NLR) dataset, a collection of 55 unique, hand-designed problems that target reasoning bottlenecks arising from the sequential prediction paradigm of LLMs and the inherently linear nature of natural language. NLR tasks require iterative updates, backtracking, and reasoning across multiple parallel chains of thought but only basic arithmetic to solve. To address these limitations, we propose a neurosymbolic reasoning approach that integrates Prolog, a symbolic reasoning engine, into the inference pipeline of LLMs. This division of labor shifts the LLM's task from iterative computations to inferring all information—explicit or implied through common sense—and encoding it as logical code. Our method yields large and robust performance gains across the GSM8k and BIG-bench Navigate benchmarks and achieves near-perfect accuracy on NLR problems, maintaining robustness even as variable interdependence—the number of other variables on which the value of a single variable depends—increases.

## 1 Introduction

The recent emergence of Large Language Models (LLMs) (Brown et al., 2020; OpenAI, 2023a; 2022; Chung et al., 2024; Meta, 2024) has revolutionized natural language processing, with LLMs demonstrating human-level performance across various professional and academic benchmarks (OpenAI, 2023b) and exhibiting an excellent understanding of linguistic rules and patterns (Mahowald et al., 2023). However, despite their linguistic competence, LLMs often demonstrate limitations in their capacity to *reason* reliably and flexibly (Mahowald et al., 2023; Dziri et al., 2024; Wu et al., 2023). These limitations likely stem from the autoregressive architecture of transformers, which enforces solving problems sequentially, constraining their ability to backtrack and recover from errors (Dziri et al., 2024). Models generate answers in a single pass of their feedforward architecture, which cannot accurately implement conditional loops (Bubeck et al., 2023). Moreover, the statistical nature of LLM training means they often fail to generalize appropriately to problems outside their training distribution (Zhang et al., 2022). Furthermore, even the most advanced LLMs, including GPT-4, have a short working memory (Bubeck et al., 2023), while reliable reasoning requires accurate and robust retrieval and integration of all relevant information.

Additionally, the linear and sequential nature of natural language contrasts with the complex, non-linear computations often involved in deductive reasoning. Even humans struggle with reasoning tasks when the brainstorming medium is confined to text. This is well illustrated by the history of logic. Aristotle's writing on syllogistic reasoning, for example, lacked the tools of symbolic logic later developed for this kind of argumentation. The result is clunky and difficult to follow, even when correct:

> If A has been proved to all or to some B, then B must belong to some A: and if A has been proved to belong to no B, then B belongs to no A. This is a different conclusion from the former. But if A does not belong to some B, it is not necessary that B should not belong to some A: for it may possibly belong to all A.

To overcome these limitations, others like Venn (1881) and Boole (1854) developed systems that allowed such reasoning to take place in a different medium—symbolic diagrams and algebraic equations, respectively.

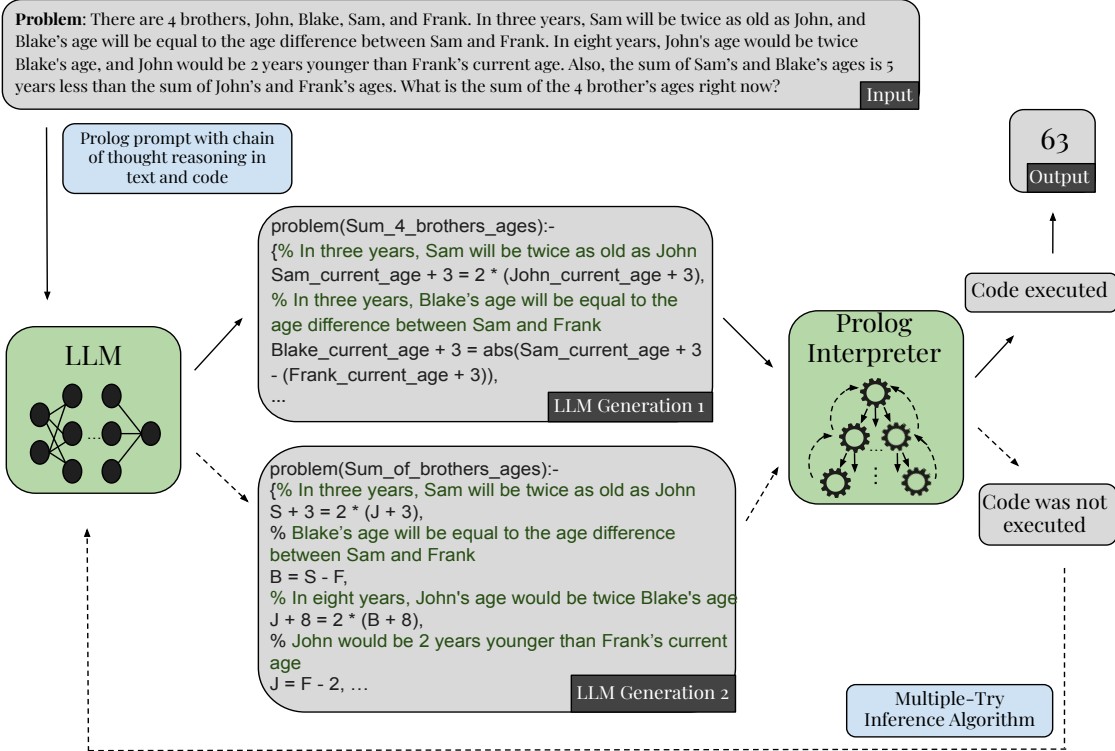

Figure 1: Our neurosymbolic approach: A natural language problem (for example, a math word problem from the NLR dataset) is given to an LLM, which is prompted to perform chain-of-thought (CoT) reasoning in text and logical code to encode the variable relationships as logical code statements. The Prolog interpreter executes the code. If the Prolog program fails, the LLM is re-prompted until valid code is generated or a limit of attempts is reached.

These tools support reasoning about much richer types of logical and causal relationships than could be easily conveyed in natural language. Jevons (1870) even developed a mechanical system for such logical reasoning, much in the spirit of Babbage's work (see Gardner (1958)). More generally, the principles and notation of mathematics allow us to concisely express concepts that would be incredibly difficult to express in natural language alone. This formalization of reasoning in systems *other* than natural language has several modern descendants, including the General Problem Solver of Newell et al. (1959), logic programming languages like Prolog (Colmerauer and Roussel, 1996), and formal tools for robust verification like Lean (mathlib Community, 2020).

In this work, we diagnose these reasoning limitations and propose a neurosymbolic solution. First, we introduce the **Non-Linear Reasoning (NLR)** dataset, a novel benchmark hand-designed to probe LLMs' reasoning shortcomings. Second, we introduce a **neurosymbolic** approach that shifts the LLM's task from deductive reasoning to mapping information in problem statements into Prolog code, effectively offloading the iterative computations involved in navigating the solution space to a reliable symbolic engine.

The NLR dataset acts as a diagnostic tool, composed of 55 unique problems across three categories. (i) **Math Word Problems** test reasoning on math problems similar to GSM8k (Cobbe et al., 2021) but with varying levels of relational complexity, particularly in their degree of variable interdependence, which we define as the number of other variables on which a single variable's value depends. (ii) **Constraint Satisfaction Problems** test the ability to trace and prune a space of possibilities based on given constraints. (iii) **Algorithmic Games** evaluate the model's ability to build and iteratively update a world model, testing working memory by requiring the model to track multiple interdependent variables.

Table 1: Examples from the NLR dataset with controlled relational/search complexity. "Variable interdependence degree $= k$" means at least one variable's value depends on $k$ other variables; the constraint satisfaction example highlights branching search from constraints that encode multiple possibilities.

| NLR Problem Statement | Complexity |
|---|---|
| **Math Word Problem**: When I was half my current age, my father was 30. When I was $\frac{1}{3}$ my current age, my mother was 25. When I was $\frac{1}{6}$ my current age, my sister was 7. If the sum of my age, my sister's age, my father's age, and my mother's age is 116, then how old am I now? | *Variable interdependence degree $= 3$*: my current age depends on {father, mother, sister}. |
| **Constraint Satisfaction Problem**: In a line to enter a cinema, 4 people are standing between Bob and Alex. Chad's index in the line is 1 after Bob's, he's standing right behind Bob considering the order of people left to right. Frank is right behind Alex. Sam is right in front of Bob. There are 2 people between Sam and Frank. If Bob is in the 7th person in the line, counting left to right, what is the number of Alex? | *Two constraints encode multiple possibilities*: e.g., both Bob→Alex and Alex→Bob orders are feasible *a priori*. |
| **Algorithmic Game Problem**: There's a cinema with 12 seats organized in 3 rows and 4 columns. Due to covid there's a policy that a seat can be filled only if none of the seats right next to it in the same column or the same row are not filled. If we place a person in the seat in the second column of the first row and then start to fill the seats left to right, row by row, starting row with 1, how many people can be seated in the cinema in total? | *Variable interdependence degree $= 4$*: each seat's availability depends on up to four neighbors (up, down, left, right). |

We show that the reasoning failures diagnosed with the NLR dataset can be alleviated by our neurosymbolic method (Figure 1). In this approach, complex relational structures can be represented accurately in symbolic form, and the LLM's task changes to understanding and extracting explicit and implicit information, a task it is proficient in. The symbolic solver (Prolog) then handles solving the systems of equations with interdependent variables, tracing all possible states, and reliably updating world models. Across GSM8k, BIG-bench Navigate (Srivastava et al., 2023), and our NLR benchmark, delegating deduction to Prolog yields large, reliable gains. On Navigate, all evaluated models exceed 98% with Prolog (vs. substantially lower with text-only CoT; see Figure 2). On NLR, GPT-4's performance increases significantly from as low as 8.3% to 100% on two of the three categories (Figure 3), and remains stable even as the degree of variable interdependence increases from 1 to 4—where text-only GPT-4's performance falls from 100% to 0% (Figure 4).

To summarize, our contributions are: (i) NLR, a 55-problem benchmark (21 math word problems, 17 constraint satisfaction, 17 algorithmic games) targeting tasks that require iterative reasoning with high variable interdependence; (ii) a neurosymbolic pipeline that integrates LLMs with Prolog for reliable reasoning; and (iii) strong empirical gains on GSM8k, BIG-bench Navigate, and the NLR dataset.

## 2 NLR Dataset

We introduce the Non-Linear Reasoning dataset (NLR), a collection of 55 problems hand-designed by the authors to probe the generality of LLMs on out-of-distribution tasks requiring iterative, non-linear reasoning while requiring only basic arithmetic calculations. The NLR dataset contains three categories of problems: (i) Math word problems (21 problems), (ii) Constraint satisfaction problems (17 problems), and (iii) **Algorithmic Games** (17 problems) (see Table 1 for examples). [1]

Unlike synthetically templated logic datasets (Han et al., 2022; Clark et al., 2020) in which all premises are stated as short clauses, NLR problems have explicit statements along with implicit information that must be inferred from common sense and context. For example, in the number-guessing problem shown in Table 2, an

---

[1]The NLR dataset is provided in the supplementary material.

implicit constraint that must be coded to arrive at the correct answer, unstated in the text, is that each digit of the number must be an integer between 0 and 9, which the model must infer.

NLR problems are designed with a controlled level of relational complexity between the variables, i.e., the *degree of variable interdependence*, which we define as the number of other variables the value of a single variable depends on. For instance, in the math word problem in Table 1, the speaker's age is defined in terms of three other people's ages (variable interdependence degree 3). Similarly, in the algorithmic problem in Table 1, seating a person affects the availability of four adjacent seats; the seat's value is interdependent with the value of its neighbors. Our experimental results show this design significantly impacts the model's ability to solve the problems end-to-end (see Section 5).

Additionally, in contrast to the MATH dataset (Hendrycks et al., 2021), which requires advanced mathematical skills (e.g., calculus, eigenvalues), in NLR we purposely confine the required operations to basic arithmetic/algebra. This separates evaluation of computational skill acquisition from reasoning, allowing us to attribute errors to models' shortcomings in tracing multiple paths to the solution, backtracking, or limited working memory rather than to missing higher-math knowledge.

**(i) Math Word Problems**   This category (21 problems) is inspired by the grade-school math problems in datasets such as GSM8k (Cobbe et al., 2021), but introduces an additional dimension of variable interdependence (i.e., higher relational complexity between the variables). These problems typically translate to systems of linear equations. We kept the total number of variables in a given problem constant (3–5 variables, comparable to GSM8k); however, the increased variable interdependence in NLR raises the number of variables co-appearing per equation. Therefore, solving the resulting systems requires multiple passes of simplification and backtracking between intermediate states of the equations. As we show in Section 5, the degree of variable interdependence has a significant effect on the performance of text-only LLMs.

**(ii) Constraint Satisfaction Problems.**   This category (17 problems) consists of finite-domain constraint satisfaction problems (Wikipedia), where a valid solution is an assignment that satisfies all constraints simultaneously. Solving such problems requires maintaining candidate sets, propagating given and derived constraints, and iteratively pruning inconsistent branches. These problems test an LLM's ability to track multiple possibilities in parallel chains of thought, backtrack between intermediate states, and reason over combinatorial search spaces—all of which are difficult to perform within the sequential next-token generation paradigm.

**(iii) Algorithmic Games**   The algorithmic games (17 problems) evaluate an LLM's ability to implement a described algorithm to track and update the states of a world model. Similar to the Navigate task (Srivastava et al., 2023), these problems provide deterministic instructions for updating an initial state. However, in contrast to Navigate, which features systematically generated instances of a single task (navigation), each problem in the NLR dataset defines a *new task* with unique rules, world models, and update logic, thus testing for generalizable reasoning.

## 3   Our Neurosymbolic Approach

To enable LLMs to perform deductive reasoning robustly, we propose integrating a reliable, deductive reasoning module into their inference pipeline. Specifically, in this study, we prompt the model to encode the constraints and relationships among variables, as described in the problem statement, as a set of Prolog code statements. The generated code is then evaluated by Prolog, which uses a deductive approach to derive a deterministic answer to the problem (Figure 1). This not only has the advantage of mirroring the likely human architecture of separate linguistic and reasoning systems (Fedorenko and Varley, 2016; Mahowald et al., 2023), but, as we show, significantly improves the performance of LLMs in mathematical reasoning.

Indeed, this approach draws on the strengths of both symbolic and neural systems. Although systems like Prolog support reliable deduction, they have no mechanism to deal with the complexities and intricacies of natural language descriptions of problems. Moreover, they are unable to perform *implicit reasoning*, which involves extracting information that is not explicitly stated in the text but is rather implied through common

Table 2: Comparison of GPT-4's CoT reasoning in text vs. in text and logical code for a constraint satisfaction problem in the NLR dataset. Red text indicates incorrect reasoning steps. Asterisks (), added manually, enclose implicit reasoning steps in the comments. In the example above, compiling the Prolog code generated by the model results in the correct answer.

**Problem:** I am a 4 digit number. My rightmost digit is not divisible by 2. The sum of my digits is 20, and all my digits are in strictly decreasing order from left to right. One of my digits is 4 times one of my other digits, and the difference between my 2 middle digits is more than 3. What number am I?

**Correct answer:** 9821

```
problem(Number):-
% The number is a 4-digit number. We can write it
    in terms of its digits:
Number #= 1000 * Digit4 + 100 * Digit3 + 10 *
    Digit2 + Digit1,
% Each of the digits of the number are single-
    digit numbers in the range [0, 9]
Digit1 #>= 0, Digit1 #< 10,
Digit2 #>= 0, Digit2 #< 10,
Digit3 #>= 0, Digit3 #< 10,
% ** Besides the leftmost digit which cannot be 0
    because otherwise the number would be a 3-
    digit number. ** The leftmost number must be
    in the range [1, 9]
Digit4 #> 0, Digit4 #< 10,
% The rightmost digit is not divisible by 2
Digit1 mod 2 #\= 0,
% The sum of the digits is 20
Digit1 + Digit2 + Digit3 + Digit4 #= 20,
% All the digits are in strictly decreasing order
    from left to right
Digit4 #> Digit3,
Digit3 #> Digit2,
Digit2 #> Digit1,
% One of the digits is 4 times one of the other
    digits. ** There are 12 possibilities for this
     condition, as there are 12 unique digit pairs
     in this number **:
(4 * Digit1 #= Digit2; 4 * Digit1 #= Digit3; 4 *
    Digit1 #= Digit4;4 * Digit2 #= Digit1;4 *
    Digit2 #= Digit3; 4 * Digit2 #= Digit4; 4 *
    Digit3 #= Digit1; 4 * Digit3 #= Digit2;4 *
    Digit3 #= Digit4;4 * Digit4 #= Digit1; 4 *
    Digit4 #= Digit2; 4 * Digit4 #= Digit3),
% The difference between the 2 middle digits is
    more than 3
abs(Digit3 - Digit2) #> 3.
```

Let's denote the four digits of the number as a, b, c, and d (from left to right).

From the problem, we know that:

1. d is not divisible by 2, which means d is an odd number. The possible values for d are 1, 3, 5, 7, 9.

2. a + b + c + d = 20, which means the sum of the four digits is 20.

3. The digits are in strictly decreasing order from left to right, which means a > b > c > d.

4. One of the digits is 4 times one of the other digits.

5. The difference between the two middle digits (b and c) is more than 3, which means |b - c| > 3.

Let's start solving the problem:

Since the digits are in strictly decreasing order and the sum of the digits is 20, a must be at least 9. If a is 9, then b + c + d = 11. Since b > c > d and d is odd, the maximum value for d is 5 (because if d is 7 or 9, then b + c would be less than 4, which is not possible because b and c must be different and greater than d).

So, if d is 5, then b + c = 6. The only possible values for b and c are 4 and 2, but the difference between b and c is not more than 3. So, d cannot be 5.

If d is 3, then b + c = 8. The possible values for b and c are 5 and 3, but the difference between b and c is not more than 3. So, d cannot be 3.

If d is 1, then b + c = 10. The possible values for b and c are 7 and 3, and the difference between b and c is more than 3. Also, one of the digits is 4 times one of the other digits (3 is 4 times 1). So, the number is 9731.

sense assumptions and context. However, Prolog and related systems excel at reasoning, with the ability to incorporate an arbitrary number of facts in their deductive processes, only generating valid conclusions given their assumptions.

Prolog expresses knowledge as a set of relations, facts, and rules, and uses a reasoning engine to run queries over these relations, applying rules through iterative resolution until a solution is found or all possibilities are exhausted. The ability to backtrack, conduct comprehensive searches, and accurately store and retrieve an arbitrary number of rules and relations are the capabilities that are difficult to implement using the feedforward architecture of LLMs, but essential for accurate deductive reasoning.

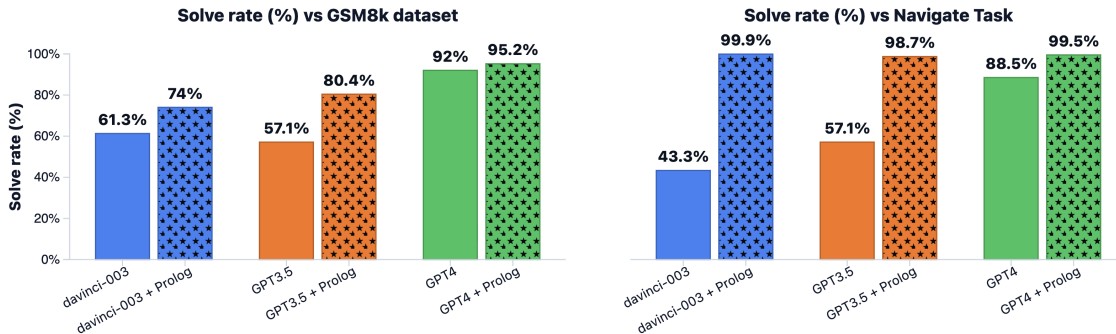

Figure 2: Comparing single-model accuracy on GSM8k and Navigate benchmarks using text-only CoT versus our neurosymbolic approach (GPT-3.5 + Prolog, GPT-4 + Prolog), using CoT in text and logical code and the Multiple-Try inference algorithm. Few-shot CoT-in-text baselines on GSM8k are reported by Bubeck et al. (2023) and OpenAI (2023b).

Moreover, in contrast to procedural or functional programming, the declarative programming paradigm of Prolog focuses on defining what to execute and the program logic rather than specifying the detailed control flow. When LLMs are prompted to generate logical code to solve a problem, this declarative nature reduces the load on the LLM to define the variables or constraints encoded in the problem in the correct order or generate all intermediate steps of the computation correctly, allowing for a more direct mapping of the information encoded in natural language statements to logical code.

Two specific design choices help this approach work well. First, we prompt the LLM to perform *chain-of-thought (CoT) (Wei et al., 2022) reasoning in text and logical code.* This in-context learning method integrates natural-language comments that walk through the implicit reasoning steps required to arrive at the intermediate variables and code statements, while the code statements encode the explicit constraints and declarative arithmetic that the Prolog interpreter needs to compile (see, e.g., Table 2). Second, we use the *Multiple-Try* inference algorithm to obtain the model's logical code generation for the problems. Using this inference method, if the Prolog code generated by the LLM fails to execute[2], we rerun the model with a slightly increased temperature (with a preset maximum number of attempts) and return the numerical answer from the model's first executable code generation (in contrast to, e.g., majority-vote schemes (Wang et al., 2022)). This approach helps to mitigate the brittleness of symbolic programming code.

## 4 Other Works

Several studies have explored integrating LLMs with symbolic reasoning modules (Olausson et al., 2023; Nye et al., 2021). A parallel line of work augments LLMs with external tools such as calculators, interpreters, and databases, which effectively decouple language understanding from computation. This separation has been shown to improve model performance on a range of logical and mathematical reasoning tasks (Chen et al., 2022; Gao et al., 2023; Schick et al., 2023; Wang et al., 2022; Zhou et al., 2022; Cobbe et al., 2021). Our work differs by specifically diagnosing and addressing reasoning failures that stem from the next-token prediction paradigm of LLMs and the inherently linear nature of natural language. We target limitations in deductive reasoning, backtracking, and handling relational complexity between variables by proposing the NLR dataset. To address these limitations, we propose a neurosymbolic approach in which the LLM constructs a symbolic world model by reasoning in both text and logical code and delegating iterative solving and state update steps to a declarative reasoning engine.

Our approach builds on and extends the work of LINC (Olausson et al., 2023) and Nye et al. (2021). LINC uses a neurosymbolic process to convert natural language into first-order logic expressions with LLMs to determine the truth value of conclusions via a symbolic theorem prover. This method has shown significant

---

[2]This primarily occurs due to variable name assignment errors, as data flows are described without mutability in Prolog's declarative syntax, unlike procedural programming.

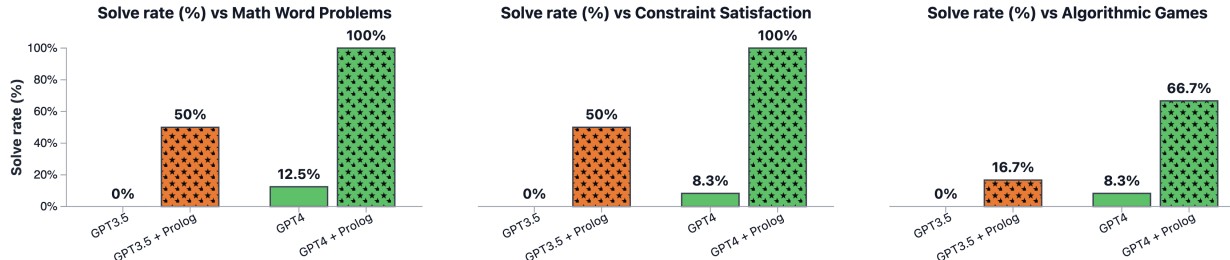

Figure 3: Comparing single-model accuracy of LLMs (GPT-3.5, GPT-4) on the NLR dataset when prompted with text-only CoT versus our neurosymbolic approach (GPT-3.5 + Prolog, GPT-4 + Prolog), using CoT in text and logical code and the Multiple-Try inference algorithm.

performance gains on the FOLIO (Han et al., 2022) and ProofWriter (Tafjord et al., 2020) datasets compared to CoT prompting. However, it has a limitation in capturing implicit information not explicitly stated in the premises, as it primarily uses LLMs as a semantic parser, translating each natural language premise directly into a logical statement (Olausson et al., 2023). Similarly, Nye et al. (2021) improve the performance of LLMs in story generation and instruction-following tasks by using a symbolic reasoning module to check the logical consistency of generated text against a minimal world model. This method increases the accuracy and robustness of neural generation but is limited by the need for hand-crafting the world model and defining specific constraints.

In our approach, the world model is constructed by the LLM itself, without any restriction on the number or complexity of clauses that can be encoded. Rather than using the LLM only as a semantic parser or text-to-code translator, we prompt it to perform chain-of-thought (CoT) reasoning simultaneously in natural language and logical code. Textual tokens embedded within the logical code enable the model to do implicit reasoning, extracting information that is not explicitly stated but must be inferred from context or common sense. This design enables a more flexible and generalizable reasoning process, making our neurosymbolic framework robust and applicable to a wider variety of problems.

Our approach is also similar to the Program-of-Thought (PoT) method (Chen et al., 2022), which separates computation from reasoning and language understanding. In PoT, LLMs generate text (as comments) and programming language statements to solve problems, delegating computation to a program interpreter. However, PoT's code statements often directly translate the comments. In contrast, our in-context prompts use comments to walk through implicit reasoning steps, making them an integral part of the CoT. This allows the comments to encode different reasoning from the code statements, extracting necessary but unstated information to generate the correct logical code (see Table 2 for examples of implicit reasoning performed by GPT-4 in the comments).

## 5 Experiments & Results

We first present results of our approach on two existing datasets: the standard mathematical reasoning dataset, GSM8k, and the Navigate task, extracted from the BIG-bench benchmark. In our experiments, we compare our approach against the standard prompting method used for reasoning tasks: examples solved using chain-of-thought (CoT) reasoning in text.

**GSM8k** GSM8k (Cobbe et al., 2021) is a widely used benchmark for mathematical reasoning tasks, comprising elementary school math problems. We tested the performance of our approach using GPT-4 and GPT-3.5 Turbo (hereafter, GPT-3.5), and text-davinci-003 on the GSM8k dataset. To construct the Prolog prompt, we selected eight problems from the first 25 problems in the shuffled test split of the dataset. This selection was made to ensure that the prompt examples covered a variety of difficulty levels (the GSM8k dataset does not provide a difficulty score for the problems). The variable names in the Prolog prompts include a brief description and the unit of the variable in order to provide additional information about the feasibility of an operation between two variables. We employed the CLP(FD) library (SWI-Prolog, a) of

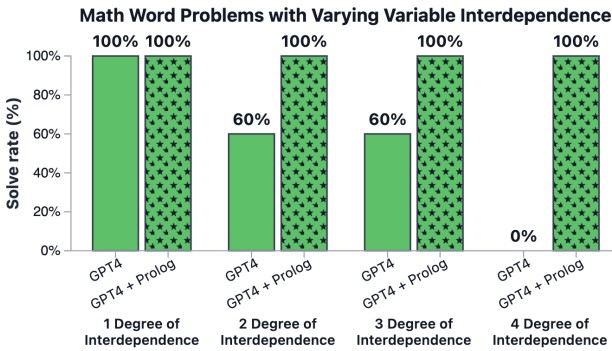 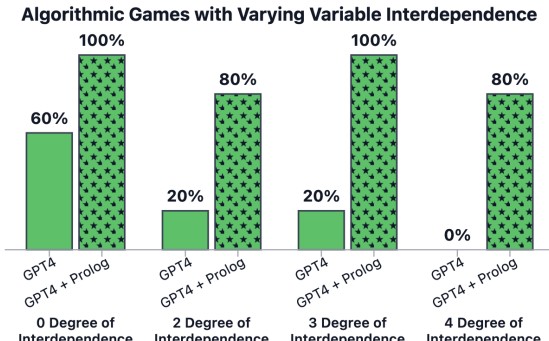

Figure 4: Comparing single-model accuracy of GPT-4 using a text-only CoT prompt versus our neurosymbolic approach on a subset of NLR problems with 0–4 degrees of variable interdependence. "$k$ degree of interdependence" means at least one variable's value depends on $k$ other variables.

SWI-Prolog to write the Prolog code for the constraint satisfaction prompts. For the math word problems and algorithmic games prompts, we used the CLP(R) library (SWI-Prolog, b). These libraries enable performing declarative arithmetic and solving systems of linear equations in Prolog.

As shown in Figure 2, integrating Prolog with text-davinci-003 and GPT-3.5 significantly improved their performance on GSM8k, highlighting the benefit of incorporating a reliable reasoning module in the inference pipeline of LLMs. The declarative nature of Prolog facilitates outlining the program logic, thus reducing the load on the LLM by eliminating the need to specify the control flow, order of operations, or accurately generate all intermediate computational steps. The performance of GPT-4 also showed considerable improvement with the integration of Prolog. It is important to note that GPT-4 is trained on the GSM8k training dataset during its pre-training phase (OpenAI, 2023b), leading to its higher performance when prompted with CoT in text.

**Navigate Dataset** The **Navigate** task is a component of the BIG-bench benchmark, a collection of tasks designed to evaluate the language understanding and generation capabilities of LLMs (Srivastava et al., 2023). This task specifically assesses LLMs' ability to solve a simple spatial reasoning task, involving tracking an agent's location based on instructions detailing the number of steps and the direction. The task requires iteratively updating a world model where each state has a few variables, the $x$ and $y$ coordinates and the direction the agent is facing.[3] Navigate, like many other benchmark datasets such as RuleTaker (Clark et al., 2020), is systematically generated. The problem statements are constructed by sampling a combination of instructions, among a pool of nine instructions, with a random number of steps added to each instruction. Originally, the Navigate task required determining whether the agent returns to its starting location, inherently introducing a 50% chance of correctness due to the binary nature of the answer. To alleviate this, we revised the task to ask the final distance of the agent from the start. To construct the Prolog prompt, we used two examples of each of the two types of problems in this dataset. Since there is no reported text-based baseline for the GPT models for the Navigate task, we prompted the models with CoT in text to establish the baseline performance for this task. For comparability, we constructed a four-shot CoT-in-text prompt with the same four problems that were used to build the Prolog prompt.

As shown in Figure 2, performance exceeds 98% for all three models integrated with Prolog on the Navigate task, a significant improvement compared to the performance of models prompted with CoT in text. This suggests that the integration of a symbolic reasoning module can help prevent arithmetic errors in LLMs and assist with the models' limited working memory when tracking and updating variables of the world model states. Notably, the text-davinci-003 and GPT-3.5 models, which have a more restricted working memory relative to GPT-4, showed the most significant improvements.

---

[3]We selected Navigate for our experiments among other BIG-bench tasks due to the fact that PaLM 540B and PaLM 62B's performance on the task were significantly below the best human performance score (Chowdhery et al., 2022).

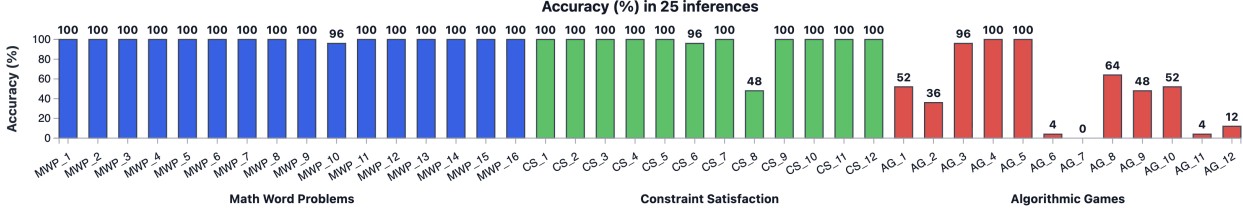

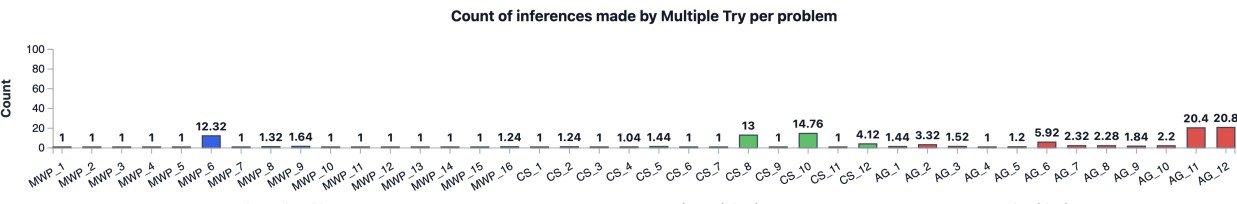

Figure 5: Evaluating the robustness and performance variability of our best model, GPT-4 + Prolog, by running it 25 times on each NLR problem and recording the accuracy and average number of attempts it took to generate valid code using the Multiple-Try inference algorithm.

**NLR Dataset** We conducted three sets of experiments on this dataset. First, we compared the mean performance of our neurosymbolic approach against the text-only CoT baseline across the main NLR dataset (Figure 3), using GPT-3.5 Turbo and GPT-4.[4] [5] We used a five-shot prompt for each problem category (both for text-only CoT and our Prolog approach). The text-only prompts were generated by zero-shot prompting GPT-4 and then manually debugging its solutions.

Second, to illustrate the significant impact of variable interdependence on model performance, we designed four new instances for five math word problems and five algorithmic games problems. These instances share the same setting, variables, and reasoning patterns as the original problems but have a varying *degree of variable interdependence* from 0 to 4.[6] As shown in Figure 4, the text-only CoT performance of GPT-4 decreases drastically as interdependence increases, falling from 100% to 0% on problems with four interdependent variables. In contrast, our neurosymbolic approach (GPT-4 + Prolog) maintains consistently high performance, demonstrating its robustness to this complexity.

Third, we assessed our best model's robustness by running it 25 times on each NLR problem, recording its accuracy and the average number of attempts required by the Multiple-Try algorithm to generate valid code (Figure 5). For these experiments, we conducted inference on a total of 40 problems (16 math word problems, 12 constraint satisfaction, and 12 algorithmic games).

**(i) Math Word Problems With Variable Interdependence** As shown in Figure 3, the simple modification of defining variables in relation to other variables significantly impacts the end-to-end performance of LLMs in solving math word problems. While GPT-4 (text-only) is generally successful in extracting the equations representing the relationships between the interdependent variables from the textual information, it struggles to solve the resulting system of linear equations (in contrast to its near-perfect performance on the GSM8k dataset). This suggests that while LLMs are adept at understanding the semantic meaning of problem statements, they struggle with the non-linear computations of solving a system of linear equations when the number of simplification and variable replacement iterations required increases, even though the mathematical scope of operations is limited to simple algebra.

---

[4]The API to the text-davinci-003 model was disabled at the time we ran the experiments on the NLR dataset.

[5]The subset of the NLR dataset with varying degrees of variable interdependence is included in the supplementary material, along with the main NLR dataset.

[6]Algorithmic games and math word problems in the original NLR dataset have 3–5 interdependent variables.

As evidenced by the 100% accuracy obtained by integrating GPT-4 with Prolog, prompting the model to define the variable relationships as equalities in Prolog and utilizing Prolog's declarative arithmetic to solve the resulting system of equations completely eliminates this failure (Figure 4) and allows the models to solve these problems with increasing relational complexity between the variables robustly. Our robustness experiments (Figure 5) confirm this: the neurosymbolic model solved nearly all math problems in all 25 attempts with very few inference retries.

**(ii) Constraint Satisfaction Problems**  Figure 3 demonstrates that LLMs mostly fail to solve the NLR constraint satisfaction problems when prompted with CoT in text. The models often overlook possibilities, hallucinate about whether a possibility satisfies all constraints, or make illogical leaps in reasoning, which in turn results in an 8% success rate in solving these problems. For instance, consider the constraint satisfaction problem presented in Table 1, which states that there are four people between Bob and Alex in a line. The LLMs only consider the possibility where Bob is standing in the $i$th position and Alex in the $(i+5)$th position of the line, which is one of the two possible orders of Bob and Alex. The existence of constraints that encode multiple possibilities makes these problems particularly difficult for LLMs due to the extensive non-linear reasoning required to trace and revisit all potential solutions.

However, as demonstrated by Figure 3, prompting GPT-4 to formulate the information encoded in the problems as Prolog predicates, where the model's task is to encode the constraints as logical code statements rather than attempting to iteratively check the possible states against the constraints, effectively addresses these issues. This approach resulted in a 100% success rate in solving the problems. GPT-3.5, on the other hand, succeeds in correctly encoding the constraints as logical code in Prolog half of the time, which demonstrates that GPT-3.5 is less proficient than GPT-4 in inferring the information implied by the natural language statements of the problems.

**(iii) Algorithmic Games**  As shown in Figure 3, GPT-4 solved 66.7% of the problems with the integration of Prolog, a significant improvement compared to the 8% success rate achieved by the model when solving these problems using text only. When tasked to solve the problems end-to-end, the models struggle to accurately store and retrieve the intermediate states, which are typically lists of length 10, often omitting numbers when rewriting the updated list, failing to consider all of the conditionals in the given algorithm during updates, or applying the algorithm for the incorrect number of steps.

While offloading to Prolog helped, Figure 5 shows this category had the lowest robustness and required the most attempts to generate valid code. This is anticipated, as these tasks require the LLM to correctly implement the data structure for initializing and updating a world model, a complex code-generation task. Thus, faithfully translating the game's rules into a valid program placed a higher load on the LLM's code-generation and implicit reasoning abilities.

## 6 Limitations

The main limitation of the NLR dataset is scalability. Designing tasks that require unique reasoning patterns for resolution—math word problems with high variable interdependence, constraint satisfaction problems with constraints encoding multiple possibilities, and game algorithms with new rules—is complex and time consuming. Moreover, LLMs may produce coherent but incorrect logical code solutions, making error detection challenging. Additionally, Prolog's limited infrastructure support for complex data structures, compared to languages like Python, may restrict its applicability to problems involving higher-dimensional data.

## 7 Conclusion

This work diagnoses and mitigates key limitations of LLMs in reliable, generalizable reasoning. We (i) introduce a hand-designed benchmark that targets iterative computation with controlled degrees of variable interdependence; (ii) propose a neurosymbolic approach where iterative deduction is delegated to a declarative solver (Prolog) and the LLM's task shifts to implicit reasoning elicited through chain-of-thought prompting in text and logical code; (iii) empirically show that this method substantially improves reasoning performance on

three different benchmarks; and (iv) demonstrate that performance of text-only models degrades sharply as the degree of variable interdependence increases in problems, revealing a bottleneck in iterative computation, while our neurosymbolic approach remains robust.

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
