# OpenReview forum: "Reliable Reasoning Beyond Natural Language"
_TMLR — Withdrawn by Authors_

### Review · Reviewer_y5ZW · 2025-11-04

**Summary Of Contributions:**

The paper is about augmenting math problems with some information from Prolog. The authors show numbers for gpt4, gpt3.5 turbo & text-davinci 003 on GSM8k, Big Bench navigate & custom dataset they introduced. As far as I know  3.5 turbo & davinci were deprecated long ago.
 They also introduce a dataset with 55 hand crafted examples.  There are 3 types of problems within these 55 problems. Navigate has 250 training instances. The test set must be smaller. GSM8k test has 1.3k which is good. However as the authors point out gpt 4 was trained on GSM-8k.
The ideas introduced are good but need a lot of more work.

**Audience:**

No

**Audience Explanation:**

The paper is vague with little evidence to convince of its importance. The authors need to provide clear evidence of the test set size and reproducibility.

**Broader Impact Concerns:**

This is just prompt engineering. Right now, there is one table (table 2) which might show what the actual prompt was. The table is confusing and vague to guess any concerns

**Claims And Evidence:**

No

**Claims Explanation:**

There are many unanswered questions:
- What is the test set size in each case? It is easy to design prompts to overfit smaller test cases.
- Please provide numbers on standard benchmark leaderboard.( https://huggingface.co/blog/open-llm-leaderboard-mmlu). Small changes in the prompt can cause huge difference in numbers
- What does the information from prolog look like

**Requested Changes:**

- Something is confusing in this line " Asterisks (), added manually, enclose implicit reasoning steps in the comments"  on page 5. The table below it is confusing. What are the 2 columns there? Please separate the actual input & the text from prolog clearly
- As noted in the paper, gpt 4 was already trained on gsm8k. With many new math benchmarks, may be you could provide numbers on one of the latest ones. Please search for latest papers on oss or gpt 5 models and choose math/reasoning datasets from it
- Openai has released many reasoning models. How does this work with reasoning models. There are many open source reasoning models too.
- Creating synthetic data with SLM or LLM is easy with the right prompt and iteration. Please look into it and extend your custom dataset.
- It would help to add few open source models as well

---

### Review · Reviewer_LpZu · 2025-11-10

**Summary Of Contributions:**

The paper contributes along three axes:
1. It proposes the "Non-Linear Reasoning" benchmark (NLR), a set of 55 tasks that assess LLM problem-solving capabilities when complex arithmetic deduction is necessary.
2. It proposes a neurosymbolic approach that combines chain-of-thought prompting with Prolog calls. The LLM attempts to convert the textual problem description into Prolog code which is then passed on to Prolog for symbolic reasoning. This is repeated iteratively until the Prolog interpreter returns a numerical answer or an attempt limit is reached.
3. It reports on a set of experimental results on the NLR as well as more established reasoning benchmarks using three LLM models. The results show that the proposed NLR is more challenging than its counterparts and that the combination with Prolog is useful across the board, boosting all of the tested LLMs' performances on all tasks.

Strengths:

a. Getting LLMs to solve text-based reasoning problems is useful for users who lack the training needed to formalize said problems in Prolog or other symbolic systems. The proposed neurosymbolic approach is a step in that direction.

b. The NLR benchmark complements the GSM8k and BIG-bench Navigate benchmarks by introducing tasks with a higher level of interdependence between key variables. This is both a realistic measure as applications may necessitate such levels of interdependence and a way to test LLMs "out of distribution" as they are typically trained on GSM8k and the like.

Weaknesses:
a. Relationship to very closely related work is not discussed. The contributions of this submission may stand on their own, but I believe a discussion of the following two papers is necessary. In particular, for Zhengkun, et al., a number of reasoning datasets are used that could be discussed in this submission; why are these datasets complementary to the proposed NLR tasks?
- Yang, Xiaocheng, Bingsen Chen, and Yik-Cheung Tam. "Arithmetic Reasoning with LLM: Prolog Generation & Permutation." Proceedings of the 2024 Conference of the North American Chapter of the Association for Computational Linguistics: Human Language Technologies (Volume 2: Short Papers). 2024.
- Di, Zhengkun, et al. "Lorp: LLM-based logical reasoning via prolog." Knowledge-Based Systems (2025): 114140.

b. Some aspects of the proposed method are unclear. For instance, on page 7, the authors write: "To construct the Prolog
prompt, we selected eight problems from the first 25 problems in the shuffled test split of the dataset.". It is unclear to me how you construct the Prolog prompts here. One or two examples of your full prompts should help clarify this.

c. The authors have ~2 pages that could be used to provide deeper insights into the experimental results. For example, on page 9, the authors state: "While GPT-4 (text-only) is generally successful in extracting the equations representing the relationships between the interdependent variables from the textual information, it struggles to solve the resulting system of linear equations (in contrast to its near-perfect performance on the GSM8k dataset).". Could an example illustrating this failure mode be added?

**Additional Comments:**

I have no ethical concerns.

**Audience:**

Yes

**Audience Explanation:**

Reasoning in LLMs is a topic of wide interest in machine learning at the moment.

**Broader Impact Concerns:**

I have no ethical concerns.

**Claims And Evidence:**

Yes

**Claims Explanation:**

To a large extent, yes. In particular, differences in task complexity between the proposed NLR and existing reasoning benchmarks are clearly explained. However, addressing Weaknesses b and c in "Summary of Contributions" would substantiate the authors' claims more strongly.

**Requested Changes:**

I would like the authors to address all three Weaknesses I have listed in "Summary of Contributions". Weaknesses a and b are critical to securing my recommendation whereas Weakness c would strength the work.

Additional minor comments:

- Figure 1: it is unclear whether 63 is the correct answer to the problem. If it is and it was produced by the first LLM call (LLM Generation 1), then the Prolog program succeeded and thus there is no need to re-prompt the LLM.

- The term "non-linear reasoning" could have other meanings in computer science, see [https://www.cs.princeton.edu/~zkincaid/_static/pub/popl18a.pdf] for example. Is there a standard definition for the term in the "LLM reasoning" space?

Typos or the like:
- p1: "professional and academic benchmarks"; I am not sure the term "professional benchmark" is meaningful. Perhaps "industrial benchmark"?
- p1: "Models generate answers in a single pass of their feedforward architecture, which cannot accurately implement conditional loops" --> "Models generate answers in a single forward pass and thus cannot accurately implement conditional loops"
- p1: "If A has been proved...belong to all A."; is this a quote from somewhere? I agree that it is a clunky and difficult to follow set of statements, but am not sure it is necessary to conveying the point you are making about limitations of syllogistic reasoning (unless it is a quote from Aristotle?)
- p2: Fig. 1 caption, "limit of attempts" --> "attempt limit"
- p4 and p10: "end-to-end" --> "end to end", see [https://english.stackexchange.com/a/586840]
- p4: in the listing of the three types of problems, only (ii) has a full stop after the name of the category; please make them consistent.

---

### Review · Reviewer_26Mm · 2025-11-15

**Summary Of Contributions:**

This paper introduces a neurosymbolic reasoning approach that integrates Prolog with LLMs to address fundamental limitations in LLM reasoning - particularly the lack of capacity to reason reliably and flexibly, due to their autoregressive nature. The authors introduce the Non-Linear Reasoning (NLR) dataset, which highlights reasoning bottlenecks in LLMs, especially regarding deductive reasoning, backtracking ability, comprehensive search, and the storage of rules and relations in working memory. In the proposed framework, the LLM extracts implicit and explicit information and generates Prolog statements and code, while being relieved of iterative computation responsibilities. The method demonstrates substantial improvements on three datasets: GSM8k, BIG-bench Navigate, and the novel NLR dataset.



Strengths:
1. The NLR dataset is novel, manually curated, and effectively designed to expose specific deficiencies in LLM reasoning capabilities
2. The experimental section clearly demonstrates the benefits of the proposed division of labor: the LLM extracts implicit and explicit information from problem statements in the form of Prolog code, while the Prolog engine handles the actual problem resolution: the gains are significant.

Weaknesses:
1. Model coverage is limited to one family (GPT-3.5, GPT-4).
2.  More clarity is needed on concrete examples of the Prolog prompts used for GSM8k and the Navigate task. In particular, how the in-context prompts are used to elicit the right extraction of information and generation of Prolog statements.
3.  Comparison with current baselines could be improved, especially with respect to Program-of-Thought.
4.  More analysis on the Multiple-Try failure (more details in the section below).
5. The size of the dataset is limited and might raise some concern about generalization and whether it can be effectively used as a diagnostic tool for the reasoning capabilities of an LLM.

**Audience:**

Yes

**Audience Explanation:**

This work brings together researchers working on reasoning and neurosymbolic AI to address some limitations on the reasoning capabilities of current LLMs.
Additionally, this work draws attention to the autoregressive nature of LLM, which is a debated topic in the AI community.

Furthermore, the dataset is made available, which can help detect deficiencies in LLM reasoning capabilities.

**Claims And Evidence:**

Yes

**Claims Explanation:**

- The claims are supported by convincing experiments: Figure 2, 3, and 4 show that this neurosymbolic approach improves performance on all the proposed datasets.
- The problem is well presented and it highlights clear deficiencies of LLMs, as pointed out by the low score when "Degree of interpendence" increases in Fig. 4.

However, baselining and claims could be improved as described below.

**Requested Changes:**

Critical for acceptance:
- Weakness #2: The paper's clarity and reproducibility would be improved by providing concrete examples of the Prolog prompts used for GSM8k and the Navigate task (similar to Table 2 for NLR).
- Weakness #3: The authors mention that their approach is similar to Program-of-Thought (PoT), but it remains unclear whether the performance gains derive from the Prolog engine itself (as opposed to Python in PoT), or the improved in-context prompts that enhance information extraction. It is not fully clear to me whether Prolog's declarative nature is essential or whether the approach would work equally well with other backends (e.g., Python).


Recommendations to strengthen the work:
- Weakness #1: The evaluation is restricted to GPT-3.5 and GPT-4. I encourage the authors to demonstrate whether these gains transfer to other model families to establish the generality of the findings. The GSM8k evaluation is further limited by the fact that GPT-4 was trained on this benchmark during pre-training.
- Weakness #4: There are cases where the Multiple-Try method requires more than 10 attempts; more analysis would clarify the reason behind this.
- General improvement: The paper might benefit from some insights by the authors on how this system (LLM + Prolog) can be integrated for real-world deployment, given that the Prolog engine is unnecessary for many practical LLM use cases. The method appears to be a specialized tool that enhances LLM reliability specifically for problems involving complex variable interdependence, rather than a general-purpose solution

---

### Note · Authors · 2025-11-26

**Comment:**

The main contribution of this paper is the introduction of controlled non‑linear reasoning problems that expose how text‑based LLMs fail as iterative reasoning demands and variable interdependence increase, and demonstrating that a neurosymbolic approach, where non‑linear computations are offloaded to a declarative solver, restores robust reasoning performance. The problems are deliberately hand‑designed to shift reasoning from linear to iterative computation, with minimal changes to the language or the mathematical complexity, making synthetic instance generation or scaling by templating not a faithful test of generalization to such reasoning pattern.

**Withdrawal Confirmation:**

I have read and agree with the venue's withdrawal policy on behalf of myself and my co-authors.